# Selenium Modulates the Allergic Response to Whey Protein in a Mouse Model for Cow’s Milk Allergy

**DOI:** 10.3390/nu13082479

**Published:** 2021-07-22

**Authors:** Xiaoli Zhao, Suzan Thijssen, Hongbing Chen, Johan Garssen, Leon M. J. Knippels, Astrid Hogenkamp

**Affiliations:** 1State Key Laboratory of Food Science and Technology, Nanchang University, Nanchang 330047, China; x.zhao1@uu.nl (X.Z.); chenhongbing@ncu.edu.cn (H.C.); 2Division of Pharmacology, Utrecht Institute for Pharmaceutical Sciences (UIPS), Utrecht University, 3584 CG Utrecht, The Netherlands; s.thijssen@uu.nl (S.T.); j.garssen@uu.nl (J.G.); leon.knippels@danone.com (L.M.J.K.); 3School of Food Science Technology, Nanchang University, Nanchang 330047, China; 4Sino-German Joint Research Institute, Nanchang University, Nanchang 330047, China; 5Danone/Nutricia Research, Global Centre of Excellence Immunology, 3584 CT Utrecht, The Netherlands

**Keywords:** seleno-l-methionine, selenium, cow’s milk allergy, mouse model, dendritic cells, T cells, mMCP-1

## Abstract

Cow’s milk allergy is a common food allergy in infants, and is associated with an increased risk of developing other allergic diseases. Dietary selenium (Se), one of the essential micronutrients for humans and animals, is an important bioelement which can influence both innate and adaptive immune responses. However, the effects of Se on food allergy are still largely unknown. In the current study it was investigated whether dietary Se supplementation can inhibit whey-induced food allergy in an animal research model. Three-week-old female C3H/HeOuJ mice were intragastrically sensitized with whey protein and cholera toxin and randomly assigned to receive a control, low, medium or high Se diet. Acute allergic symptoms, allergen specific immunoglobulin (Ig) E levels and mast cell degranulation were determined upon whey challenge. Body temperature was significantly higher in mice that received the medium Se diet 60 min after the oral challenge with whey compared to the positive control group, which is indicative of impaired anaphylaxis. This was accompanied by reductions in antigen-specific immunoglobulins and reduced levels of mouse mast cell protease-1 (mMCP-1). This study demonstrates that oral Se supplementation may modulate allergic responses to whey by decreasing specific antibody responses and mMCP-1 release.

## 1. Introduction

Cow’s milk allergy (CMA) is one of the most common food allergies, affecting 2–3% of all infants [1]. In IgE-mediated CMA, about 85% of patients mostly experience mild symptoms, while 15% can develop severe allergic responses and about 9% develop anaphylaxis within minutes to hours after ingestion of cow’s milk [2,3]. Although 60–75% of children with IgE-mediated CMA spontaneously develop tolerance to cow’s milk before their fifth year of life, the risk of developing other atopic disorders later in life, such as asthma, is increased [4]. At the moment, no curative treatment is available for CMA and the only way to prevent allergic reactions is to avoid the intake of cow’s milk proteins. Therefore, novel approaches to prevent or treat CMA are urgently needed.

CMA is defined as an immunologically mediated adverse reaction to cow’s milk protein [5,6], which can result from a breakdown or a delay in the development of oral tolerance [7]. The major allergens in cow’s milk are αS1-casein from casein, and α-lactalbumin and β-lactoglobulin from whey. These proteins can be degraded in the intestines and pass through the epithelial barrier, after which they are presented to naïve T-cells by antigen-presenting cells such as dendritic cells (DC). In CMA, these T-cells differentiate into T helper 2 cells (Th2) which can drive allergic immune responses, including the expansion of eosinophils and mast cells, as well as isotype switch in B cells towards IgE production [8].

A number of food allergen immunotherapies are under investigation; however, these are limited in their ability to restore immune tolerance to food allergens, and often result in high rates of allergic side effects [9]. Nutritional interventions including omega-3 polyunsaturated fatty acids, prebiotics, probiotics, and symbiotic supplementation as well as Vitamin D have gathered more interest in the prevention, secondary prevention and treatment of food allergies [10,11]. With the accumulating knowledge of the involvement of micronutrients in immunological processes, there is increasing interest in the relationship between minerals and the development of immune responses. It is known that deficiencies in several minerals such as iron, zinc, copper and selenium (Se) may result in impaired and/or less efficient immune function but little is known about their specific role in food allergy development [6,12,13].

Se is a metalloid mineral which is considered to have significant potential to influence the immune system. For example, Se supplementation reduced inflammation in a Staphylococcus aureus-infected mouse mastitis model [14,15], and an increase in Se intake improved antigen-responsiveness in immune cells from adults [16]. Se has been demonstrated to improve the activation of chicken DCs in vitro [17], and dietary Se supplementation has been shown to favor differentiation of naive cluster of differentiation (CD)4+ T lymphocytes towards T helper 1 cells (Th1) [18,19], supporting the acute cellular immune response. In addition, dietary Se increased the percentage of splenic regulatory T cells (Treg) [20]. Furthermore, in a mouse model for systemic lupus erythematosus, Se supplementation inhibited activation, differentiation and maturation of B cells and macrophages [21]. Lower serum levels of selenium have also been associated with several skin diseases including atopic dermatitis [22]. People suffering from allergic asthma have significantly lower Se concentrations in their blood than healthy individuals [23].

Se status is known to affect the function of the immune cells and their ability to respond to antigens [24], which suggests that Se may affect allergic disease. Selenomethionine (SeMet) is an organic form of Se, an essential trace element that functions in the regulation of the immune response [25,26]. Oral SeMet administration suppressed ovalbumin-induced allergic sensitization in a mouse model for active cutaneous anaphylaxis by lowering Th2 cytokine production and augmenting Th1 cytokine production [27]. However, it is not known how SeMet affects CMA. Since evidence indicates that organic Se, SeMet, is better absorbed by humans and animals and has a higher bioavailability than inorganic forms of Se [28], it was investigated how CMA is modulated by SeMet-enriched diets.

## 2. Materials and Methods

### 2.1. Mice

Four-week-old, specific-pathogen-free female C3H/HeOuJ mice weaned at the age of 3 weeks were purchased from Charles River laboratory (Sulzfeld, Germany). The animals had been bred for 2 generations on a cow’s milk-free diet. Animals were housed in an individually ventilated cage (*n* = 3–6 mice/cage) at the animal facility of Utrecht University (Utrecht, The Netherlands), with a 12 h dark/12 h light cycle and ad libitum access to food and sterile water. Upon arrival, mice were randomly allocated to a negative control group (*n* = 4, the PBS group), a tolerance group (*n* = 6), a positive control group or one of the 3 Se intervention groups (*n* = 10/group). In the medium Se intervention group one mice unexpectedly died, which was not due to experimental procedures. All experiments were approved by the Animal Ethics Committee of Utrecht University (approval AVD225002016521).

### 2.2. Experimental Design

Figure 1 shows a schematic representation of the experimental design. Depending on the experimental group, upon arriving in our animal facility mice received the control diet or the diet supplemented with different concentrations of Se. The PBS group, tolerance group and positive control group were fed the cow’s milk protein-free control diet (AIN93G, obtained from sniff Spezialdiäten GmbH, Soest, Germany) which contained 0.025 mg/kg Se. SeMet (Sigma-Aldrich, St. Louis, MO, USA) was added during the manufacturing of the experimental AIN93G-based diets to obtain the following concentrations: 0.033 mg/kg Se (low Se), 0.066 mg/kg Se (medium Se) and 0.1 mg/kg Se (high Se).

During the tolerance induction phase (day −7 to −2), mice in the tolerance group were administered 50 mg whey (aWPC60, Milei, Friesland Campina, Zaltbommel, The Netherlands) in 0.5 mL phosphate buffered saline (PBS) by daily gavage dosing six times. All the other experimental groups were administered 0.5 mL PBS as a control. 

In the sensitization phase (day 0–28), mice in the PBS group were administered 10 µg of cholera toxin as an adjuvant (CT; List Biological Laboratories, Campbell, CA, USA) in 0.5 mL PBS via oral gavage. Mice in the other experimental groups were administered 20 mg whey together with 10 μg CT/0.5 mL PBS. 

On day 35, the acute allergic skin response was measured. Isoflurane-anesthetized mice received an intradermal (i.d.) injection of 10 µg homogenized whey in 20 µL PBS in the pinnae of both ears. Ear thickness was measured in duplicate for each ear after i.d. injection with whey, using a digital micrometer (Mitutoyo, Kanagawa, Japan). Whey-induced ear swelling, expressed as delta (Δ) µm, was calculated by subtracting the mean basal ear thickness before i.d. challenge from the ear thickness measured at 1 h after i.d. injection. Anaphylactic shock was monitored 15, 30 and 60 min after i.d. challenge. Anaphylactic shock symptoms were scored using a validated, previously described 0- to 4-point scoring system [29]. The body temperature was monitored at 30 and 60 min, after the i.d. challenge using a rectal thermometer. On the same day, the mice were orally challenged by gavage with 50 mg whey dissolved in 0.5 mL PBS. Eighteen hours after the oral challenge (Day 36), the mice were bled by orbital extraction under isoflurane anesthesia, followed by cervical dislocation and sectioning. All measurements were performed in a blinded manner.

### 2.3. Measurement of Allergen-Specific Immunoglobulins and Mouse Mast Cell Proteases-1 (mMCP-1) in Serum

Whey- and β-lactoglobulin (BLAC)-specific immunoglobulins (Ig) were measured in serum as described previously [30]. Briefly, high-binding Costar 9018 plates were coated with 20 µg/mL whey in a carbonate buffer or 20 µg/mL BLAC (Sigma-Aldrich, St. Louis, MO, USA) in a bicarbonate buffer and incubated overnight at 4 °C. Plates were then washed and blocked with PBS/1%BSA for 1 h. Hereafter, serum samples were incubated for 2 h at room temperature, washed, and incubated for 1 h with biotinylated rat anti-mouse IgE, IgG1 or IgG2a detection antibody (1 µg/mL; BD Biosciences, San Jose, CA, USA). The plates were subsequently washed and incubated in the dark for 45 min with streptavidin–horseradish peroxidase (HRP) (0.5 μg/mL; Sanquin, Amsterdam, The Netherlands), washed, and developed using tetramethylbenzidine (Sigma-Aldrich, St. Luis, MO, USA). The reaction was stopped with 1 M H_2_SO_4_ and absorbance was measured at 450 nm using a GloMax^®^ Discover microplate reader (GloMax, Veenendaal, The Netherlands). Whey- and BLAC-specific IgE, IgG1 or IgG2a levels are expressed in arbitrary units, which were calculated based on a titration curve of pooled sera serving as an internal standard.

Concentrations of mMCP-1 in serum were measured using a MCPT-1 (mMCP-1) Elisa kit (eBioscience, Breda, The Netherlands) according to the manufacturer’s instructions.

### 2.4. Flow Cytometric Analysis of Immune Cells

Freshly isolated single cells were obtained from spleen and mesenteric lymph nodes (MLN). Briefly, single-cell splenocyte suspensions were obtained by passing spleen samples through a 70 µm nylon cell strainer using a syringe. The splenocyte suspension was rinsed with a RPMI 1640 medium (Lonza, Basel, Switzerland) and incubated with a lysis buffer (eBioscience, Thermo Fisher Scientific, CA, USA) to remove red blood cells. The reaction was stopped by adding a RPMI 1640 medium supplemented with 10% heat-inactivated fetal bovine serum (FBS; Bodinco, Alkmaar, The Netherlands), penicillin (100 U/mL)/streptomycin (100 µg/mL; Sigma-Aldrich) and β-mercaptoethanol (20 µmol/L; Thermo Fisher Scientific, Waltham, MA, USA). Splenocytes were subsequently resuspended in this culture medium. MLN single cells were obtained by passing spleen samples through a 70 µm nylon cell strainer using a syringe and subsequently resuspended in this culture medium. Cells were counted by Coulter counter (Beckman Coulter, Beckman Coulter Life Sciences, Brea, CA, USA).

Spleen- and MLN-derived single-cell suspensions were incubated with anti-mouse CD16/CD32 (Mouse BD Fc Block; BD Pharmingen, San Jose, CA, USA) for 15 min on ice to block non-specific binding sites. Subsequently, cells were extracellularly stained with DC markers CD11c-PerCp-Cy5.5, Major Histocompatibility Complex (MHC)II-APC, CD40-FITC and CD86-PE-cy7; Th1 markers CD4- BV510, CD69-PE and CXCR3-PE-cyc7; Th2 markers CD4-BV510, CD69-PE and T1/ST2-FITC; Treg markers CD4-BV510, CD25-PerCp-Cy5.5 and CD127-PE-cyc7; and Th17 markers CD4-BV510 and CD196-PE for 45 min at 4 °C, and Treg-Foxp3-FITC and Th17-RorgT-Alexa Fluor647 (eBioscience, Thermo Fisher Scientific, CA, USA) were used for intracellular staining. Viable cells were distinguished by a fixable viability dye eFluor^®^ 780 (eBioscience, Thermo Fisher Scientific, CA, USA) and measured using a BD FACSCanto II flow cytometer (Becton Dickinson, Franklin Lakes, NJ, USA) and analyzed with FlowLogic software (Inivai Technologies, Mentone, VIC, Australia).

### 2.5. Cytokine Measurement after Ex Vivo Antigen-Specific Stimulation of Splenocytes 

For the ex vivo antigen-specific restimulation assay, spleen- and MLN-derived single-cell suspensions (5 × 10^5^ cells/well) were cultured in U-bottom culture plates (Greiner, Frickenhausen, Germany) with either medium, whey (500 µg/mL) or anti-CD3 (1 µg/mL) at 37 °C, 5% CO_2_. Culture supernatants of anti-CD3 stimulated cells were collected after 48 h, and supernatants of whey-stimulated cells after 5 days. Samples were stored at −20 °C until further analysis. Levels of IL-10, IL-17A, IL-13, IL-4, IFN-γ and TNF-α were analyzed using a customized Luminex kit, according to manufacturer’s instructions.

### 2.6. Statistical Analysis

Data are presented as mean ± standard error of the mean (SEM), including individual data points. Data between groups were compared using parametric one-way analysis of variance (ANOVA) with nonparametric Kruskal–Wallis test with Dunn’s post hoc test. All statistical analyses were performed using GraphPad Prism software (version 7.03; GraphPad Software, San Diego, CA, USA) and results were considered statistically significant when *p* < 0.05.

## 3. Results

### 3.1. Acute Allergic Skin Response and Body Temperature after Challenge 

To assess the effect of dietary Se supplementation on the clinical response in the allergic animals, the mice were challenged i.d. with whey protein after which the acute allergic skin response (ASR), shock score and body temperature were measured. No significant difference was observed in shock score (Appendix A). Although the ear swelling in the medium Se-supplemented animals (Δ ear swelling 102 ± 22 µm) was not significantly different from the mice in the tolerance group, no significant effects of Se supplementation could be observed when these groups were compared to the positive control (Δ ear swelling 122.8 ± 25µm) (Figure 2A).

Similarly, 30 and 60 min after the i.d. challenge in both ears, a drop in mean body temperature was observed in the sensitized animals that had not received whey in the tolerance induction phase, but only the mice in the positive control group had a significantly lower body temperature when compared to the PBS and the tolerance group (Figure 2C,D). At 30 min after the i.d. challenge, no significant effects of Se supplementation on body temperature could be observed when these groups were compared to the positive control (Figure 2C). Interestingly, 60 min after the i.d. challenge mean body temperatures in the low and high Se groups (36.8 ± 12.5 °C and 36.7 ± 2.0 °C, respectively) were not significantly different when compared to the positive control group (35.6 ± 2.9 °C). In contrast, the mean body temperature in the medium Se intervention group was significantly higher (37.7 ± 0.6 °C) than the allergic positive control group (Figure 2C), indicating a protective effect of Se, although not in a dose-dependent fashion.

### 3.2. Dietary Se Supplementation Downregulates Whey- and BLAC-Specific Immunoglobulin Levels 

To assess whether Se influences allergic sensitization, whey- and BLAC-specific IgE, IgG1 and IgG2a were measured in serum. Whey- and BLAC-specific IgE and IgG subclasses were significantly increased in the positive control group compared to the PBS group (Figure 3). A clear suppression of the specific antibody responses was observed in the tolerance group, as none of the allergen-specific immunoglobulin levels were found to differ from those measured in the PBS control group. Whey-specific IgE was significantly lower in the medium Se group compared to the positive control group (Figure 3A), and BLAC-specific IgE was significantly lower in the high Se group compared to the positive control group (Figure 3D). Whey- and BLAC-specific-IgG1 levels were not significantly affected by dietary Se supplementation (Figure 3B,E). However, significantly lower levels of whey- and BLAC-specific-IgG2a were observed in all Se intervention groups when compared to the positive control group (Figure 3C,F).

### 3.3. Dietary Se Supplementation Reduces mMCP-1 Production

To assess the influence of dietary Se supplementation on mast cell degranulation, the concentrations of serum mMCP-1 were measured (Figure 4). Compared to the tolerance group, the mean serum concentration of mMCP-1 was significantly higher in the positive control group. In Se-supplemented groups, mMCP-1 levels appeared to follow a concentration-dependent downward trend, although this only reached significance in the highest Se group.

### 3.4. Dietary Se Supplementation Affects Dendritic Cell Activation 

The effect of Se supplementation on the activation status of splenic DCs was assessed in freshly isolated splenocytes by means of flow cytometry, the gating strategy of splenic DC is shown in Appendix A. The percentage of CD11c+MHCII+DCs was significantly higher in the positive control group compared to the PBS and tolerance group (Figure 5A). However, Se supplementation did not significantly affect the percentage of CD11c+MHCII+DCs when compared to the positive control group. Similarly, CD11c+MHCII+DCs in the positive control group were found to have a significantly higher expression of CD40 and CD86 when compared to the PBS group, but no significant differences in CD40 expression were observed when the Se-supplemented groups were compared to the positive control group (Figure 5B,C). However, the expression of CD86 in CD11c+MHCII+DCs was significantly lower in the medium Se group when compared to positive control group (Figure 5C).

### 3.5. Dietary Se Only Modestly Affects T Cell Differentiation

To determine whether specific differentiation of T-cell subsets was modulated by Se, freshly isolated splenocytes and MLNs were analyzed by flow cytometry (Figure 6). In the spleens of animals in the positive control group, the percentage of Th2 was significantly higher when compared to the PBS group, but no significant effect of Se supplementation was observed (Figure 6A). In the MLN, although the percentage of Th2 cells was higher in the medium Se group compared to the low Se group, no difference was detected when the positive control group was compared to the PBS and the tolerance group (Figure 6B). In addition, we also investigated percentages of activated Th2 cells, activated Th1 cells, Tregs and Th17 cells. However, because in many cases outcomes in the negative control group did not significantly differ from the positive control group, it was not possible to assess the influence of dietary Se supplementation on the cell populations (data shown in Appendix A).

### 3.6. Cytokine Production after Ex Vivo Stimulation with Whey

Cytokine production was measured to determine the effect of dietary Se supplementation on functional responses of splenocytes and MLN cells upon ex vivo exposure to whey and anti-CD3. Overall, no differences in IL-17A, IL-13, IFN-gamma or TNF-alpha in Se intervention groups were observed compared with the positive control group (Appendix A). However, whey-specific IL-4 production was significantly higher in the positive control group compared to the PBS and tolerance groups, and it was significantly lower in the medium Se intervention group when compared to the positive control group (Figure 7A). Similarly, a significantly higher whey-specific production of IL-10 was observed in the positive control group when compared to the PBS group (Figure 7B). Interestingly, when compared to the positive control group, whey-specific IL-10 production was significantly higher in the medium Se group. In anti-CD3 stimulated cells, Se supplementation did not affect IL-4 and IL-10 production when compared to the positive control group (Appendix A). Ex vivo stimulation of MLN cells with either whey or anti-CD3 did not induce detectable cytokine production (data not shown).

## 4. Discussion

The present study demonstrates that Se supplementation can affect various immune parameters in a murine model for whey-induced food allergy. Although most clinical outcomes appeared to be unaffected, a significant effect on body temperature (anaphylaxis readout) was observed in the mice receiving the medium Se diet. These results may be partially explained by and associated with the effects of Se supplementation on antibody responses.

An increase in allergen-specific IgE, IgG1 and IgG2a, such as those observed in the current study, is typical for IgE-mediated CMA [30,31]. Dietary supplementation with Se led to lower serum levels of whey- and BLAC-specific IgG2a, which appeared to be independent of the concentration of Se. Concentrations of BLAC-specific IgE were significantly affected only in the high Se group, and whey-specific IgE levels were observed to be significantly lower in the medium Se group. In a murine model for systemic lupus erythematosus, oral administration of Se reduced the production of disease-associated autoantibodies [18]. Moreover, the latter study demonstrated that Se treatment of activated B-cells led to a concentration-dependent decrease in the expression of CD69, CD80 and CD86 on B cells. In the current study, we did not include the measurement of B-cell activation markers, but we were able to detect a modest but significant reduction in the expression of the costimulatory molecule CD86 in DCs isolated from mice in the medium Se group. Se has previously been reported to enhance surface molecule expression of DCs and accelerate their differentiation and maturation [17,32], but as those observations were shown in in vitro studies it is not clear what could account for these conflicting data. DCs can take up, transport, process and present antigens to T cells [33], and interaction with highly activated DCs will favor priming of the T-cells, whereas low activated DCs will induce tolerance [34]. CD86 expression is reportedly upregulated in food allergy in mice [35], and lowering the expression of this molecule in DCs may thus affect allergen specific T-cell activation. In turn, this could impair the ability of the T-cells to activate B-cells, leading to lower allergen-specific antibodies. However, this remains speculative as we did not measure B-cell activation markers. In a murine model for atopic dermatitis, oral administration of Se suppressed the development of atopic dermatitis-like skin lesions, lowered total IgE levels, reduced skin expression of IL-4 and led to a lower number of mast cells in the skin [36]. In our current study, we observed a significantly lower expression of serum mMCP-1 in mice that were fed the high Se diet, suggesting that mast cells may be affected also by this dietary intervention with Se. In line with this result, it has previously been shown that mast cell degranulation was significantly decreased by exposure of MC/9 cells, a mast cell line, to Se in vitro [37]. In the latter study, it was found that β-hexosaminidase and histamine release were reduced, which is in line with our in vivo observation of reduced mMCP-1 levels. In addition, IL-4 expression by splenocytes derived from mice in the medium Se group was significantly lower. IL-4 plays a critical role in the development of Th2 cells and subsequent allergic reactions. IL-4 drives the differentiation of Th2 cells, which also secret IL-4, thereby inducing a positive feedback to reinforce Th2 differentiation [38]. Although we did not observe any prominent effects of dietary Se supplementation on Th2 cell populations, the difference in IL-4 expression might also be linked to mast cells which can also produce IL-4 upon IgE-mediated stimulation [39]. A reduction in the expression of IL-4 could potentially reinforce anti-allergic effects of Se supplementation, as degranulation of mast cells which have been shown to develop from progenitor cells present in spleen during an antigen-specific T-cell response [40] may be enhanced by IL-4. 

A decline in IL-4 expression levels may signal an increase in mast cell differentiation, as IL-4 is known to induce apoptosis in developing mast cells [41]. However, we demonstrated that allergen specific IL-10 expression was enhanced in the splenocytes derived from mice in the low and medium Se groups. IL-10 can potentially inhibit cell survival of developing mast cells [42] and has pleiotropic anti-inflammatory effects [43], such as the downregulation of IL-4 production by Th2 cells [44]. The exact source of the IL-10 in our in vivo culture is not clear, but it could potentially be derived from CD5+ B cells, which have been shown to inhibit antigen-mediated activation of mast cells in vitro as well as allergic responses in mice in an IL-10-dependent manner [45]. Whether specific subsets of B-cells are differentially affected by Se remains to be investigated.

During oral sensitization and challenge with the allergen, the intestinal immune cells are the first line to encounter the administered antigen. Food allergy may already be initiated in the MLN, as removal of these lymph nodes prior to sensitization results in reduced food allergic reactions [46]. In contrast to those observations, we observed significantly lower percentages of Th1 cells in the MLN of the positive control mice, but percentages of Th2 cells did not differ from those found in the PBS group. Although we did find several modest but significant effects (e.g., a higher Th1/Th2 ratio and a lower percentage of activated Th2 cells) when the positive control group was compared to the medium Se group, we cannot draw definite conclusions from these findings with regards to the role of Se in food allergy. Similarly, the T-cell populations in the spleens of positive control mice were mostly similar to those of the PBS group, the only exception being a significantly lower percentage of Th1 cells. Previous studies on mice have reported that a high Se diet skewed the Th1/Th2 balance towards a Th1 phenotype in the spleen [18], but compared to the positive control group in our study, Se supplementation appeared to have the opposite effect. Furthermore, Treg cells are known to play important roles in modulating immune responses, exerting anti-inflammatory effects and inducing oral tolerance [47]. Se suppressed inflammation in chronic colitis mice by upregulating CD4+CD25+ Treg cells [48]. However, in our experiment there were no differences to be detected in Treg cell populations, which might be due to the dose of Se intake as well as Se form. Taken together, our data suggest that within the current experiment splenic and MLN-derived T-cell populations remained mostly unaffected by Se intervention in allergic sensitization. Whereas in other studies decreasing Th1, Th17 and increasing CD4+CD25+Treg cell responses were found [20,48,49].

In summary, this current study shows that Se supplementation can affect some clinical aspects associated with food allergy. Clear effects on T cell populations were not detectable. It might be that Se supplementation modulates allergic responses to whey by decreasing the specific antibody responses and mMCP-1 release. In addition, according to the result of mMCP-1, Se may affect allergic responses by affecting mast cell activation and degranulation in a direct way. Understanding how Se specifically affects the immune system may lead to better use and understanding of Se-containing supplements aimed at allergy management.

## Figures and Tables

**Figure 1 nutrients-13-02479-f001:**
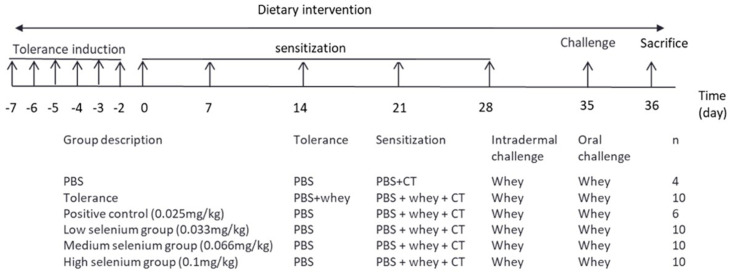
Schematic representation of the experimental model and the experimental groups. Challenge at day 35 was given i.d. but also intragastrically by gavage dosing. At day 36, upon sacrifice of the mice, blood, spleen and mesenteric lymph nodes (MLN) were collected for further analysis. CT: cholera toxin; PBS: phosphate buffered saline.

**Figure 2 nutrients-13-02479-f002:**
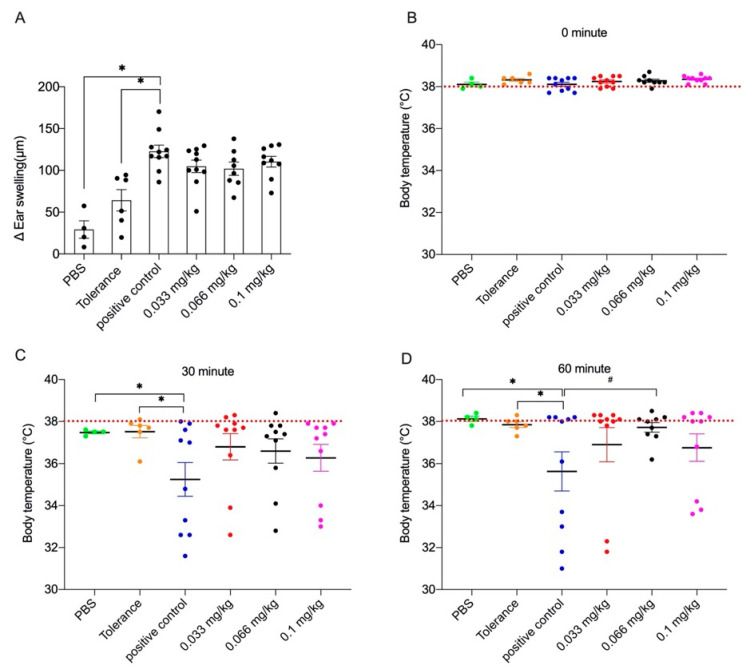
The acute allergic skin response and body temperature after i.d. challenge in control and whey-sensitized mice fed with the control or Se diets (day 35, PBS group *n* = 4, tolerance group *n* = 6, positive control group or low and high Se intervention groups *n* = 10/group, medium Se intervention group *n* = 9). Ear swelling was measured before and 1 h after i.d. challenge with whey (**A**). Body temperature (**B**–**D**) was measured before and 30 as well as 60 min after i.d. challenge. Values are expressed as mean ± standard error of the mean (SEM) including individual data points. Significant differences between PBS, tolerance and whey-sensitized mice are indicated by * *p*  <  0.05. Differences between whey-sensitized mice fed the control diet and those fed the Se diets are indicated by # *p*  < 0 .05. Differences are analyzed with a one-way analysis of variance (ANOVA) followed by a Kruskal–Wallis non-parametric test. i.d.: intradermal.

**Figure 3 nutrients-13-02479-f003:**
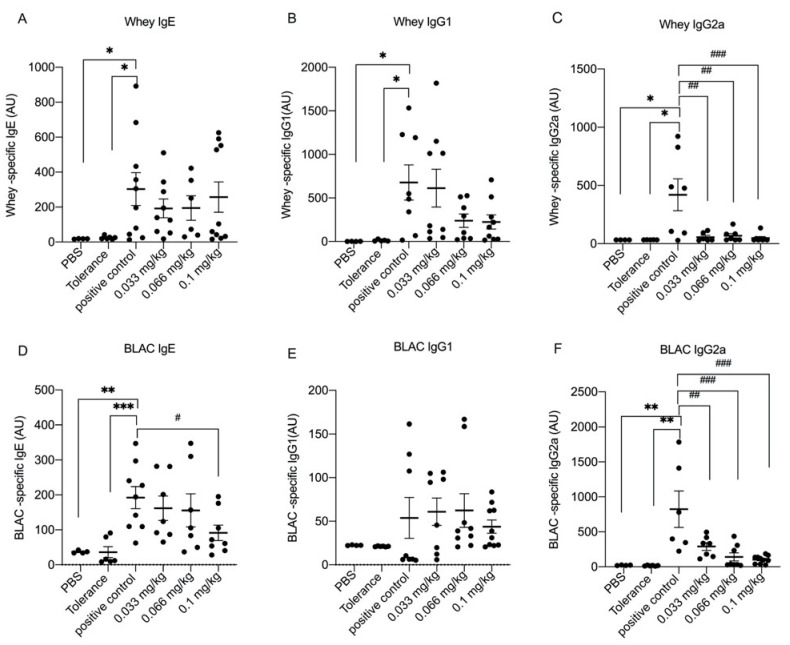
Serum levels of whey-specific immunoglobulins (Ig) after the oral challenge with whey in control and whey-sensitized mice fed the control or Se-based diets. Serum (PBS group *n* = 4, tolerance group *n* = 6, positive control group or low and high Se intervention groups *n* = 10/group, medium Se intervention group *n* = 9) was harvested 18 h after the oral challenge with whey. Serum levels (day 36) are shown in arbitrary units (AU) for (**A**) whey-specific IgE, (**B**) whey-specific IgG1, (**C**) whey-specific IgG2a, (**D**) BLAC-specific IgE, (**E**) BLAC-specific IgG1 and (**F**) BLAC-specific IgG2a. Values are expressed as mean ± SEM including individual data points. Significant differences between PBS, tolerance and whey-sensitized mice are indicated by * *p*  < 0 .05, ** *p*  <  0.01 and *** *p*  <  0.001. Differences between whey-sensitized mice fed the control diet and those fed the Se diets are indicated by # *p*  < 0 .05, ## *p*  < 0 .01 and ### *p*  <  0.001. Differences are analyzed with a one-way analysis of variance (ANOVA) followed by a Kruskal–Wallis non-parametric test. Ig: immunoglobulins, BLAC: β-lactoglobulin.

**Figure 4 nutrients-13-02479-f004:**
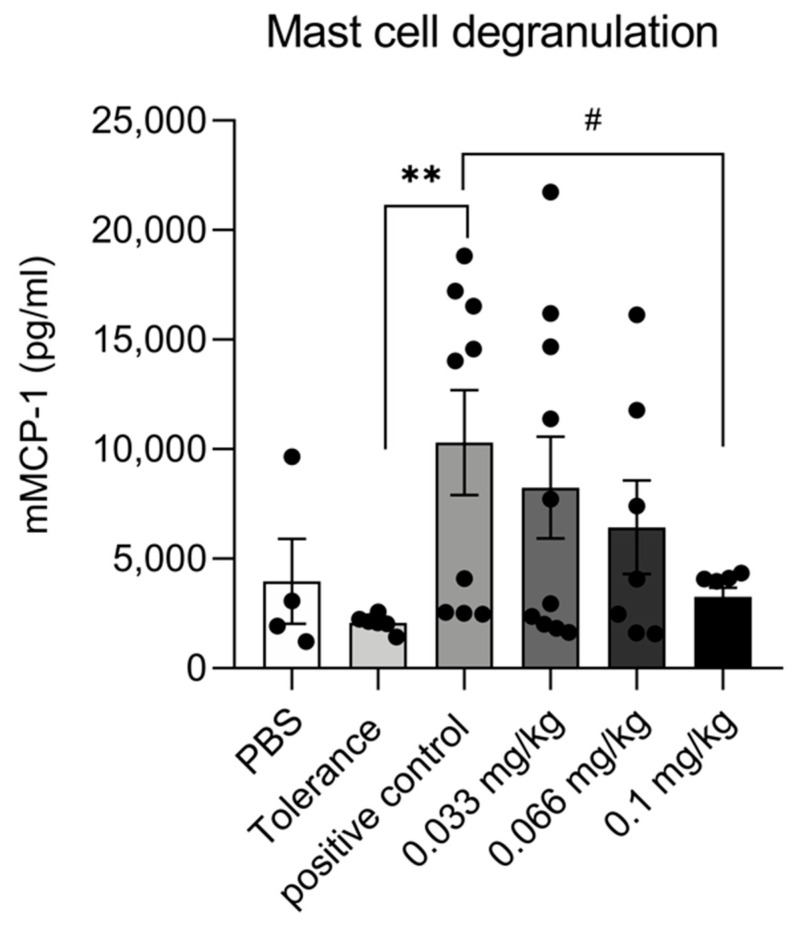
Serum levels (day 36) of mouse mast cell protease-1 (mMCP-1) after the oral challenge with whey in control and whey-sensitized mice fed the control diet or Se-based diets (PBS group *n* = 4, tolerance group *n* = 6, positive control group *n* = 9, low Se intervention group *n* = 10, medium and high Se intervention groups *n* = 7/group. We were unable to collect enough serum from one mouse in the positive control group and two mice in the medium and high Se intervention groups). Data are expressed as pg/mL including individual data points. Significant differences between PBS, tolerance and whey-sensitized mice are indicated by ** *p* < 0.01. Differences between whey-sensitized mice fed the control diet and those fed the Se diets are indicated by # *p* < 0 .05. Differences are analyzed with a one-way analysis of variance (ANOVA) followed by a Kruskal–Wallis non-parametric test.

**Figure 5 nutrients-13-02479-f005:**
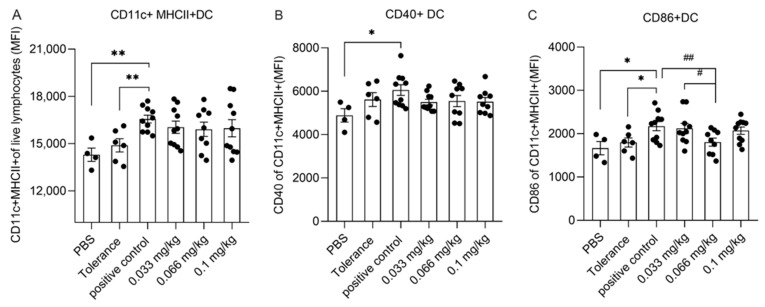
Flow cytometric analyses of dendritic cells (DCs) and surface markers expression in the spleen at day 36. Spleen DCs were analyzed based on their expression of CD11c and MHC-II. Median fluorescence intensity of CD11c+ MHCII+ DCs was under the gate of single and live cells in total splenocytes (**A**), activation status was further distinguished based on their MFI of surface markers CD86 (**B**) and CD40 (**C**). PBS group *n* = 4, tolerance group *n* = 6, positive control group or low and high Se intervention groups *n* = 10/group and medium Se intervention group *n* = 9. Significant differences between PBS, tolerance and whey-sensitized mice are indicated by * *p*  <  0.05 and ** *p*  < 0 .01. Differences between whey-sensitized mice fed with the control diet and those fed with the Se diets are indicated by # *p*  < 0 .05, ## *p*  <  0.01. Differences are analyzed with a one-way analysis of variance (ANOVA) followed by a Kruskal–Wallis non-parametric test.

**Figure 6 nutrients-13-02479-f006:**
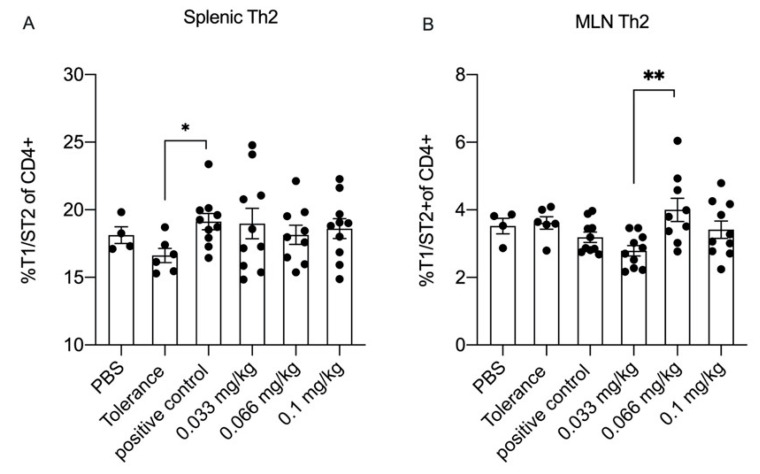
Flow cytometric analysis of T helper (Th) 2 in Spleen and mesenteric lymph nodes (MLN) at day 36. Th2 cells were defined as T1/ST2+ of CD4+ cells. Percentages of splenic Th2-cells (**A**), percentages of MLN Th2-cells (**B**). Data are presented as mean  ±  SEM of PBS group *n* = 4, tolerance group *n* = 6, positive control group or low and high Se intervention groups *n* = 10/group and medium Se intervention group *n* = 9 including individual data points; significant differences between PBS, tolerance and whey-sensitized mice are indicated by * *p*  <  0.05 and ** *p*  < 0 .01. Differences between whey-sensitized mice fed with the control diet and those fed with the Se diets are indicated by # *p* < 0 .05 and ## *p*  <  0.01. Differences are analyzed with a one-way analysis of variance (ANOVA) followed by a Kruskal–Wallis non-parametric test.

**Figure 7 nutrients-13-02479-f007:**
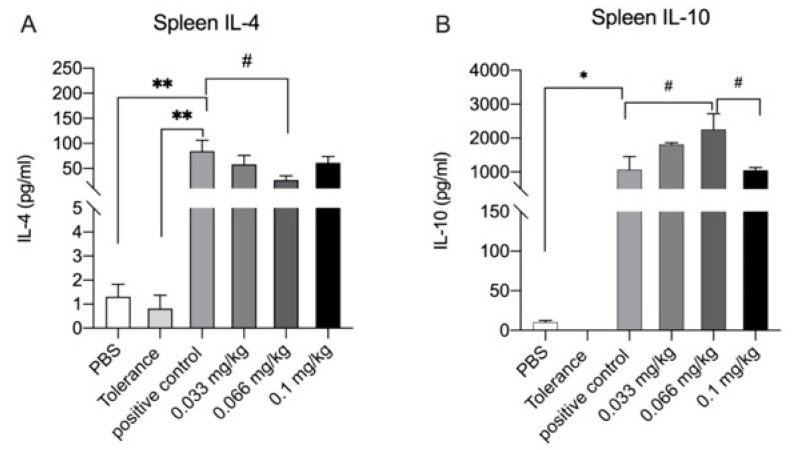
Cytokine concentrations (pg/mL) were measured in supernatant of ex vivo stimulated splenocytes for 5 days with whey. (**A**) IL-4, (**B**) IL-10. Data are presented as mean ± SEM. Significant differences between PBS, tolerance and whey-sensitized mice are indicated by * *p*  <  0.05 and ** *p*  < 0 .01. Differences between whey-sensitized mice fed the control diet and those fed the Se diets are indicated by # *p*  < 0 .05. Differences are analyzed with a one-way analysis of variance (ANOVA) followed by a Kruskal–Wallis non-parametric test.

## Data Availability

The authors will send detailed data and calculations on request.

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
