# Peer review of "Selenium Modulates the Allergic Response to Whey Protein in a Mouse Model for Cow’s Milk Allergy"

_nutrients, 2021, doi:10.3390/nu13082479_

Round 1

Reviewer 1 Report

The manuscript "Selenium modulates the allergic response to whey protein in a mouse model for cow’s milk allergy" is a scientifically sound and interesting study. I have found that some parts of the text are difficult to follow, though, and I recommend that they are rewritten in a simpler way. Also, the description of material and methods should be improved in terms of clarity at some points. Please, find my specific comments below:

  • Abstract: Line 18-Mice were intragastricalLY sensitised. Please, correct accordingly.
  • Figure 1: Line 101- Were mice sensitised intragastrically or orally? Why is it said intragastrically in the abstract? Why were mice challenged both i.d and orally?
  • Line 145: Why only MLNs (draining the gut) were isolated and not skin draining lymph nodes? Is there any skin immune response known to be associated with Se?
  • Results: Line 182-192- This paragraph is difficult to follow and information does not come to the reader. I suggest focusing on describing the results of Se supplementation. The rest is shown in the figures and graphs.
  • Line 232- Significantly with no capital letter.
  • Figure 4: It seems that the n is lower than stated in M&M for some experimental groups. Why is that? Please indicate the "n" in every figure caption.
  • Line 270-290: Same as above. Try to simplify and rewrite this paragraph as the information given gets diluted and the reader does not receive a straight message.
  • Figure 6: Please, increase the font size for each graph. I would simplify the figure including only Th2 graphs and add the rest of graphs to supplementary material.
  • How do you explain that most effects on immune responses were observed with the medium Se dose? How were Se concentrations determined?

Reviewer 2 Report

The standout results in he paper are the reduced IgG2a antibody responses and the reduced serum mMCP‐1 after oral challenge. The slightly reduced IgE response and slightly less temperature reduction at the 0.066mg/kg dose are barely significant with respect to both the controls and the other Se doses. The reduced serum mMCP‐1 does not correlate with the reduced IgE and temperature drop since it clearly has a dose response gradient. Most importantly the abstract should be changed since it suggests there is clear evidence of important IgE reductions and IgE was reductions and biological consequences.

Selenium has been shown reduce mast cell mediator release. (Influence of selenium on mast cell mediator release (Ref 38). The possibility this affected the mMCP‐1 results should be better discussed.

The reduced IgG2a is particularly important since references 18-20 are cited to show that in other systems the Se supplementation increases Th1 responses. It would be worthy of follow up.

There is no mention of reproducing the experiments and I consider it would be important to show the results of at least a second experiment to show that the middle 0.066 dose does have more effect than the higher and low doses on several measurements. The large variation in the experimental groups compared to the controls is an obvious problem.

It would be appreciated if the method section spelt out the markers for the Th1, Th2, Th17 and Treg

Figure 3. IU should be AU for antibody titration graphs

The study starts with very young  mice (4 weeks) and does not give information on when they were weaned and when the selenium supplementation was started. Measurements of selenium should have been performed. Perhaps I have missed it.

Author Response

Please see in the attachment.

Round 2

Reviewer 2 Report

NIL